

# Climatic control on seasonal variations of glacier surface velocity

Ugo Nanni[1,2], Dirk Scherler[3,4], Francois Ayoub[5], Romain Millan[6], Frederic Herman[7], Jean-Philippe Avouac[1]

[1]Division of Geological and Planetary Science, California Institute of Technology, Pasadena, CA, USA.
[2]now at University of Oslo, Department of Geosciences, Norway.
[3]Earth Surface Geochemistry, GFZ German Research Centre for Geosciences, Potsdam, Germany.
[4]Institute of Geographical Sciences, Freie Universität Berlin, Berlin, Germany.
[5]Jet Propulsion Laboratory, California Institute of Technology, Pasadena, CA, USA.
[6]University of Grenoble Alpes, CNRS, IRD, IGE, Grenoble, France.
[7]Institute of Earth Surface Dynamics, University of Lausanne, 1015 Lausanne, Switzerland.

*Correspondence to*: Ugo Nanni (nanni@uio.no)

**Key Points:**

1. We retrieve 10-day changes in glacier velocity over 7 years for 48 glaciers in the Western Pamir.

2. Glaciers accelerate in spring (on 38 glaciers) and in autumn (on 24 glaciers).

3. Accelerations appear controlled by meltwater and shed light on rapid glacier response to changes in air temperature.





**Abstract**

Accurate measurements of ice flow are essential to predict future changes in glaciers and ice caps. Glacier displacement can in principle be measured at the large-scale by cross-correlation of satellite images. At weekly to monthly scales, the expected displacement is often of the same order noise for the commonly used satellite images, which limits the retrieval of accurate glacier velocity. Assessments of velocity changes on short time scales and over complex areas such as mountain ranges are therefore still lacking, but are essential to better understand how glacier dynamics are driven by internal and external factors. In this study, we take advantage of the wide availability and redundancy of satellite imagery over the Western Pamir to retrieve 10-day glacier velocity changes over 7 years for a wide range of glacier geometry and dynamics. Our results reveal strong seasonal trends. In spring/summer, we observe velocity increases of up to 300% compared to a slow winter period. These accelerations clearly migrate upglacier throughout the melt-season, which we link to changes in subglacial hydrology efficiency. In autumn, we observe glacier accelerations that have rarely been observed before. These episodes are primarily confined to the upper ablation zone with a clear downglacier migration. We suggest that they result from glacier instabilities caused by sudden subglacial pressurization in response to (1) supraglacial pond drainage and/or (2) gradual closure of the hydrological system. Our 10-day resolved measurements allow us to characterize the short-term response of glacier to changing meteorological and climatic conditions.

**Short summary**

Glacier movement can be measured with satellite images. Over weeks to months, this movement is often too little compared to noise. We analyze thousands of images and retrieve, for 7 years, 10-day velocity changes over 48 glaciers in the Pamir. We capture their responses to rapid meteorological changes. In spring, strong glacier accelerations propagate upglacier. In autumn, accelerations propagate downglacier. Both result from changes in meltwater input to the glacier bed.



## 1 Introduction

Glaciers and ice caps are shrinking around the world in response to current global warming
caused by human activities (IPCC, 2021). Projecting their future changes is critical to assess the
associated impacts in terms of sea-level-rise, naturals hazards and water resources (Azam et al., 2021).
Such projections rely on our ability to observe and understand glaciological processes at various
temporal and spatial scales. A key manifestation of glacier dynamics is surface velocity (hereinafter
referred to as "velocity" for brevity), which depends on both internal (e.g., glacier geometry, bed types)
and external (e.g., air temperature, precipitation) factors. Although field-based measurements can
provide very accurate data about glacier processes (e.g., Stevens et al., 2022; Nanni et al., 2022a;
Vincent et al., 2022), they require human presence and are generally limited to accessible areas,
resulting in poor spatial coverage. In contrast, satellite imagery provides wide spatial coverage and is
therefore an effective approach for retrieving glacier-flow characteristics in remote areas and on a large
or even global scale (Li et al., 1998; Berthier et al., 2005; Scherler et al., 2008; Rignot et al., 2011;
Dehecq et al., 2015, 2019; Gardner et al., 2019; Millan et al., 2022).

When using satellite imagery to study glacier flow, cross-correlation techniques are often used
to track the movement of features on the glacier surface (Scambos et al., 1992; Kääb et al., 2002;
Leprince et al., 2008; Heid and Kääb, 2012). Obtaining an accurate glacier velocity depends on the
ability to obtain a high signal-to-noise ratio (SNR) for the expected glacier displacement (meters to
kilometers per year). The SNR depends on the methods chosen and the quality of the images (e.g.,
resolution, georeferencing, orthorectification) used for cross-correlation (Heid and Kääb, 2012; Millan
et al., 2019). The time interval between images that are correlated is an important parameter, as a larger
time interval will increase the signal (i.e., the displacement) relative to the noise of the measurement.
Annual to multi-year time periods allow expected displacements to be large relative to the noise, so that
automated procedures can effectively recover accurate velocity variations on global (e.g., ice sheets,
mountain ranges) and local (e.g., a small mountain glacier) scales (Quincey et al., 2009; Rignot et al.,
2011; Scherler et al., 2011b; Dehecq et al., 2015; Mouginot et al., 2017)



These observations, when coupled with appropriate modelling and complementary datasets (e.g., ice thickness), allow monitoring of surface-mass balance changes over long periods (several years) and assessing glacier response to climate change (Tedstone et al., 2013; Kjeldsen et al., 2015; Dehecq et al., 2019). On seasonal timescales, velocity variations have been documented on outlet glaciers and large mountain glaciers, where displacements are large enough to be well-above the noise level (Scherler and Strecker., 2012; Armstrong et al., 2017; Altena and Kääb, 2017; Usman and Furuya, 2018; Derkacheva et al., 2020; Minchew and Joughin, 2021; Yang et al., 2022; Beaud et al., 2022). Various processes are responsible for seasonal variations of glacier dynamics, as well as glacier surges and flow instabilities, and probably result from hydrological controls (Tedstone et al., 2013; Moon et al., 2014; Quincey et al., 2015; Stearns and Van Der Veen, 2018; Derkacheva et al., 2021; Bouchayer et al., 2022). On shorter timescales (weeks to months), the expected displacement is often on the same order as the noise for commonly used satellite imagery (Millan et al., 2019), which limits the retrieval of accurate glacier velocities. Most studies that have looked at changes in glacier flow between weeks and months tend to focus on individual glaciers (Scherler and Strecker, 2012; Armstrong et al., 2016; Derkacheva et al., 2020; Riel et al., 2021; Beaud et al., 2022), and only a few studies have conducted such investigations over larger areas (Altena et al., 2019, Yang et al., 2022), demonstrating the opportunity to better understand how glacier dynamics (e.g., basal sliding) and glacier instability are affected by climatic conditions (Tedstone et al., 2013; Stearns and Van Der Veen, 2018; Kääb et al., 2018; Beaud et al., 2022).

New generations of medium-resolution, short-recurrence-time optical sensors, such as Landsat-8 (15-m resolution in the panchromatic band, 16-day recurrence time, operational since 2013) and Sentinel-2 (10-m resolution in the visible bands, 5-day recurrence time, operational since 2016), have fostered new methodological developments to measure glacier velocity variations on relatively short (< month) time scales (Armstrong et al., 2017; Altena et al., 2019; Derkacheva et al., 2020; Riel et al., 2021). When using these medium-resolution images, it is crucial to preserve the true signal while removing poor correlations during post-processing, preferably in an automated manner to be applied over large areas. Recent studies have proposed efficient and accurate use of the large database of available satellite imagery to retrieve accurate short-term changes in glacier velocity (Altena et al.,





2019; Derkacheva et al., 2020; Riel et al., 2021; Hippert-Ferrer et al., 2021). However, most of these studies have focused on fast outlet glaciers (Derkacheva et al., 2020; Riel et al., 2021) or other fast glaciers with large monthly-scale velocity changes (Altena et al., 2019; Millan et al., 2019; Hippert-Ferrer et al., 2021). It is not yet clear whether these approaches can be used to study the short-term dynamics of mountain glaciers in general, or only for some of the largest and fastest glaciers. Large-scale assessments of velocity changes on short time scales and over complex areas such as mountain ranges are therefore still lacking.

In this study, we examine short-term (weekly to monthly) changes in glacier velocity in the western Pamir, a mountain range that hosts a wide variety of glaciers in terms of geometry and dynamics. Our study aims to assess the temporal and spatial extent to which significant changes in glacier velocity can be recovered using the latest generation of optical sensors with global coverage. We first present a semi-automated processing protocol designed to examine changes in velocity on weekly to monthly time scales over entire mountain ranges. We then apply our methodology in the Western Pamir region to investigate its performance, as this region features glaciers with different characteristics and geometries as well as low cloud cover. We finish by discussing the observed velocity variations and the ability of our approach to provide information on the interactions between meteorological conditions and glacier dynamics.

## 2 Study site

The Western Pamir in Tajikistan (Figure 1) is characterized by high peaks, steep slopes and deep narrow valleys that are home to abundant mountain glaciers, of which the 72-km long Fedchenko Glacier is the largest mountain glacier outside the polar regions. The regional equilibrium line altitude (ELA) lies between 4600 and 4800 m asl (Aizen, 2011). Average glacier velocities have been stable in this region since 2008 and the glaciers have been interpreted to be close to equilibrium by Dehecq et al. (2019). The climate is characterized by westerlies-controlled snowfall during winter and warm and dry summers (Yao et al., 2012). Long-term temperature measurements at the Goburnov station (elevation of





4200 m, close to the Fedchenko Glacier) show positive daily mean air temperatures from late May to early October (Lambretch et al., 2014, 2018; Figure S1). We focus our study on seasonal variations in

130    glacier velocity that repeat from year to year and do not investigate surges or inter-annual dynamics.

From an area of 60 km x 60 km that is approximately centered on the Fedchenko Glacier, we investigated 48 glaciers from which we selected 38 glaciers (lines in Fig. 1c) for analysis of seasonal velocity variations. This selection contains a wide range of glaciers (see Fig. 1 caption for abbreviations): from small (< 5 km long) and narrow (< 500 m wide; e.g., glacier H765), to mid-sized

135    (e.g., glacier RGS) and very large glaciers (> 20 km long), such as GG and FED (Figure 1 d to g).

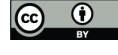

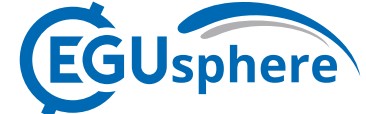



**Figure 1 .** *(a) Glacier velocity (m.d⁻¹) mosaic of the Western Pamir mountain range area, averaged over the time period 2013-2020. Velocity is obtained at a horizontal resolution of 40 m using 650 pairs of*



*Sentinel-2 images and 400 pairs of Landsat-8 images (see Material and Methods for details). Colorbar is coded on a linear scale going from black (no displacement) to white (fast displacement). Text boxes show the names (based on Randolph Glacier Inventory Consortium, (2017)) of the 7 glaciers investigated in detail: Fedchenko Glacier (FED); Tanymas 3 Glacier (T3); Grumm-Grzhimaylo Glacier (GG); Russian Geographical society Glacier (RGS); Walter 731 Glacier (W731); Hadyrsha 765 Glacier (H765); Malyy Tanymas Glacier (MT). All of these glaciers flow northward, except for the RGS that flows southward. (b) Location of the Western Pamir in the regional context; blue areas show glacier coverage (Randolph Glacier Inventory Consortium, (2017)); background elevation provided by ASTER (2019).'(c) Glaciated area (gray), flowlines (green) of the 7 glaciers previously mentioned and the flowlines (dark grey) of 31 additional glaciers investigated. (d, e, f, g) Characteristics of the studied 38 glaciers, in terms of median velocity, median elevation, length and median width. Easting and Northing are relative to 38.39N 72.63E in the UTM 32N projection.*

## 3 Materials and methods

### 3.1 Image selection and cross correlation

We downloaded Landsat 8 and Sentinel 2 images that cover our study area for the time periods above from USGS EarthExplorer (https://earthexplorer.usgs.gov/, last access: September 9th 2022) and Copernicus Scihub (https://scihub.copernicus.eu, last access: September 9th 2022), respectively. As we are interested in short-term changes in glacier velocity, we correlated images that are separated by 16 and 32 days for Landsat 8 and of 10, 20 and 30 days for Sentinel-2. These repeat cycles provide good temporal coverage while maintaining sufficient accuracy for measuring the expected glacier velocities in our region (Lambrecht et al., 2014; Millan et al., 2019). To minimize residual stereoscopic effects, we only correlated image pairs from the same sensor and tile (see Fig. S2 for the number of correlations per cycle and sensor). We used panchromatic band (B8, 15-m spatial resolution) imagery of the USGS/NASA Landsat 8 data set for the years 2013-2020, and band 8 (10-m spatial resolution) imagery of the ESA Sentinel-2 data set for the years 2017-2020 (Figures S2, S3 and S4 in the supporting information). These images are georeferenced and orthorectified with quality assessment bands (for Landsat 8), which contain information on surface, atmospheric and sensor conditions. Glacier contours





provided by the Randolph Glacier Inventory (RGI Consortium, 2017) are used to distinguish between glaciers and stable ground.

The displacement that occurs between the acquisition of two images is measured using ENVI's COSI-Corr add-on software (freely available at 170 http://www.tectonics.caltech.edu/slip_history/spot_coseis/, last access: September 9th 2022). We used a frequency correlation with a search window size ($w$) that ranged from 64 x 64 pixels to 32 x 32 pixels. Note that by apodising with a 2D Hanning function, the spatial resolution is better than the correlation window size (Leprince et al., 2007). This technique had already been used on mountain glaciers and yielded a stated 1σ uncertainty of about 1/10 of the pixel size (Scherler et al., 2008; Herman et al., 175 2011). The displacement is calculated in steps ($s$) of 4 pixel in the x- and y-directions, resulting in a displacement ground sampling distance of 60 m for Landsat-8 and 40 m for Sentinel-2. Steps smaller than the search window size allow for measurement redundancy (Leprince et al., 2008), which increases its accuracy compared to steps that are similar to the search window (Fahnestock et al., 2016). The output of COSI-Corr consists of three images, the East-West ($EW$) and North-South ($NS$) components of 180 the displacement and the associated Signal-to-Noise Ratio ($SNR$), which assesses the quality of the correlation at each pixel. The results of all correlations are combined into a single data cube for each sensor before applying our post-processing procedure.

### 3.2 Basic filtering

We applied a basic filtering procedure to the $NS$ and $EW$ components of each surface 185 displacement that does not depend on the calculated total displacement. We started by setting a threshold for the signal to noise ratio ($SNR>=0.97$) to exclude poor correlation results (Scherler et al., 2008). In addition, we used the quality assessment band for Landsat 8 images to flag pixels where clouds, including cirrus, are present in either one of the correlated images. After calculating the total displacement amplitude (herineafter referred to as the velocity), we removed pixels associated with 190 unrealistically high velocity values that exceed a threshold value of 15 m.d$^{-1}$ (5475 m.yr$^{-1}$). The chosen value of this threshold was guided by previous estimates of annual glacier velocities (Dehecq et al., 2015), but set sufficiently high to allow for physically meaningful variations. Finally, to avoid edge



effects, we removed a band as wide as the correlation window size from the edge of the image in all displacement maps.

### 3.3 Advanced filtering

#### 3.3.1 Temporal redundancy of the velocity

We calculated the median magnitude of the *NS* and *EW* components ($NS_{med}$, $EW_{med}$) of the velocity and the associated median absolute deviation (MAD) and then removed all pixels where: $|NS - NS_{med}| > 10 * NS_{MAD}$, or $|EW - EW_{med}| > 10 * EW_{MAD}$. This threshold is defined based on previous estimates of annual glacier velocities in the area (e.g. Dehecq et al. (2015)). In doing so, we assume temporal coherence of the glacier velocity over time, which has the advantage of eliminating potential miscorrelations (Scherler et al., 2008) but at the risk of eliminating episodes during which glacier velocity changes dramatically. Our choice of threshold is motivated by the fact that we focus on short-term changes in glacier velocity rather than surge events, although the above filter can be easily adjusted to avoid exclusion of surge events. Our approach is simpler than the ones proposed by Derkacheva et al. (2020) and Riel et al. (2021), in which velocity time series are interpolated and filtered with multiple local regressions.

#### 3.3.2 Spatial redundancy of the velocity

We assessed the validity of the measurement at each pixel by examining the similarity with neighboring values in a search area centered on the pixel in question. The measurement redundancy due to correlation depends on the window size *w* and the correlation step *s* and has a size of $\frac{2w}{s}$ pixels (Leprince et al., 2007). This redundancy represents the distance to be covered to observe an independently recorded displacement. The search area is defined as a centred square with a half-length of $1 + \frac{2w}{s}$ pixels, which means that half of the pixels are expected to have the same value within this square. Each pixel for which the surrounding square does not have at least half of its values similar (within one standard deviation) to that of the queried pixel is removed, both for the *EW* and *NS* components. This filtering procedure is similar to a spatial median filtering but based solely on the





characteristics of the COSI-Corr correlation process and it does not make any assumptions about the glacier flow (Leprince et al., 2007).

### 3.4 Calibration and stable ground correction

We further applied a filter to remove the bias introduced by uncorrected satellite attitude variations during orthorectification. To do this, we used non-glacial areas as defined by the RGI, assuming they represent stable ground, calculated the average velocity in the along-track and across-track directions of the *EW* and *NS* velocity and removed them from the corresponding velocity maps. This procedure follows that of the COSI-Corr "destriping" tool, which has been shown to improve the accuracy of velocity measurements (Scherler et al., 2008). We then estimated remaining bias from the distribution of the velocity measured over the non-glacial areas. We fitted the distribution of these measurements with a Gaussian curve using the least squares criterion and subtracted for each individual pair the Gaussian mean so that the distributions of the *EW* and *NS* velocity values were centered on zero. Finally, we removed isolated measurements, i.e., pixels surrounded by less than three non-zero values.



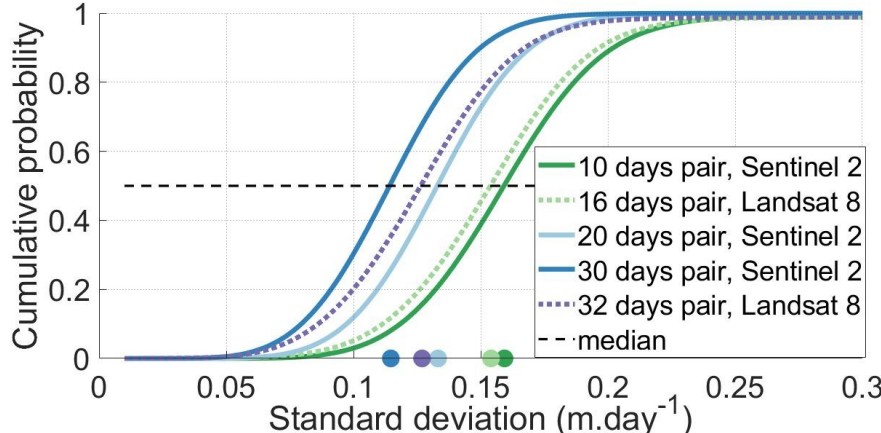

**Figure 2** . *Cumulative distribution of the standard deviation of velocity in ice-free areas as a function of repeat cycle length for Sentinel-2 and Landsat 8 data. Coloured points show the median value of the distribution for each repeat cycle, which represents the nominal precision achieved for each cycle.*

**3.5 Measurement precision**

To assess the capability of each sensor to map short-term changes in velocity, we calculated the cumulative distribution of the standard deviation of the velocity in ice-free areas for different repeat cycle lengths (Figure 2). We did not use the standard deviation in glaciated areas, as it likely represents natural fluctuation of velocity rather than the sensor precision (Figure 1). The distribution is calculated for each pixel of all measured velocities. We assume that the median value of the distribution represents

the nominal precision achieved for each cycle. Such analysis of the standard deviation has been shown to reliably quantify sensor precision in mountain areas (Millan et al., 2019). We observe that the precision on velocity measurements used in our study ranges from 0.11 to 0.16 m.d$^{-1}$ and decreases, as expected, with time separation between two acquisitions. For Sentinel-2 we obtain a precision of 0.16, 0.13 and 0.11 m.d$^{-1}$ for 10-, 20- and 30-day cycles, respectively. For Landsat 8, we obtain a precision of

0.15 and 0.12 m.d$^{-1}$ for 16- and 32-day cycles, respectively. Similar results were achieved in the study by (Millan et al., 2019). Note that the nominal precision obtained for Landsat 8 and Sentinel-2 is very similar. Assuming that the nominal precision depends only on image correlation, we calculated the sub-pixel matching precision as $\frac{\sigma_{cycle} c}{ps}$, where $\sigma_{cycle}$is the standard deviation of a given cycle, $c$ is the cycle




length and *ps* the pixel size. We find a sub-pixel image matching precision of 0.15 and 0.33 pixels for a
10-day cycle with Sentinel 2 and for a 32-day cycle with Landsat 8 imagery respectively. We get back
to the measurement precision in the Discussion section.

### 3.6 Retrieving velocity variations along the glacier flowline

We calculated the velocity magnitude as the norm of the NS and EW components. We then
calculated the median velocity over the entire time period at each pixel location and used the resulting
map (Figure 1) to define the central glacier flowline. For each correlation result, we extracted the
velocity profile along the flowline by taking the median value from a 3x3 pixel square. In order to
calculate a regularized time series over the period 2013-2019, we defined a time grid with a temporal
resolution of 20 days. We chose this resolution because it represents the average time between pairs
over the study period (Figure S4). For each 20-day time step, we searched for velocity maps where the
center of the time range falls in the 20-day period and calculated the median value of all available
velocity estimates from different sensors and repeat cycles. This median-based approach has proven to
be statistically robust when the sampling of the time series is sufficiently dense (Derkacheva et al.,
2021), which is the case in our study. We did not use a weighted median approach as we found that it
introduced more noise than a regular median, which could be due to the fact that our repeat cycles are of
similar length (i.e. 10 to 32 days). We also calculated a 10-day annual velocity for each glacier by
averaging the flowline profiles over a year using the same approach as described above. The higher
resolution is made possible by the large number of image pairs available for each 10-day time step
(Figure S4).

### 3.7 Terminology

In the following, we refer to changes in velocity with respect to the annual median velocity. We
express these changes both in absolute values (m.d$^{-1}$) and in relative values (%); i.e., a value of -10%
corresponds to a velocity 10% below the median velocity. We use the term 'acceleration' to refer to
abrupt changes in the velocity pattern, when the glacier velocity increases -typically- by more than 35%.
Such accelerations can have different onset date at different elevation. We then consider any period of





the year in which the velocity is typically 35% above (below) the annual median velocity, a time of faster (slower) velocities.

## 4 Results

We analyzed 48 glaciers in the Western Pamir, from which we selected 38 that show clear
seasonal changes in velocity over the 7-year period investigated (Figure 1c, green and black flowlines). We first present a detailed analysis for a selection of 7 of these glaciers, which we consider to be representative for different geometries and dynamics, and then we present results for the full selection.

### 4.1 Seasonal velocity variations of the Fedchenko Glacier

We show in Fig. 3a the 20-day changes in the velocity calculated along the Fedchenko Glacier
flowline (Figure 1, FED) from spring 2013 to winter 2019-2020. We observe repeating seasonal variations in glacier velocity of up to 1 m.d$^{-1}$ (Figure 3a), which corresponds to changes of up to 150% near the glacier front (Figure 3c). Such changes are higher than those described in the study of Lambrecht et al. (2014). In the ablation area (> 20 km downglacier distance), we observe pronounced changes relative to the median velocity with negative changes (-0.2 to -0.4 m.d$^{-1}$) towards the end of
each year (September to December) and positive changes (0.3 to 0.7 m.d$^{-1}$) around the second quarter of each year (March to July). Seasonal variations in the accumulation area are less pronounced, partly because the SNR is lower, which is probably related to areas of slow velocity and low-texture due to snow cover (Figure 3b). These seasonal variations are observed throughout the 7 years (Figure 3a), but are more accurately represented during 2018 and 2019, likely because of the combination of Landsat 8
and Sentinel-2 data.

Stacking velocities of all individual years into a characteristic 10-day annual velocity time series removes some of the noise from individual years (Figure 3c, d) and shows that the annual evolution of velocity changes follows a characteristic pattern, best described by an upglacier migration of glacier acceleration. At the end of February, the lowest part of the ablation zone (> 50 km downglacier
distance), where the median velocity is about 0.25 m.d$^{-1}$, accelerates by 0.3 m.d$^{-1}$, i.e. more than 100 %.



Between April and May, the acceleration extends for about 10 km upglacier (45 km downglacier distance) with similar changes, and simultaneously, the previously fast flowing areas near the terminus start to slow down. By the end of June, the acceleration has migrated c. 35 km upglacier (20 km downglacier distance), where the median velocity is about 0.5 m.d$^{-1}$, reaching changes in velocity by more than 0.7 m.d$^{-1}$, or 50 to 150 %. In June and July, velocities in the upper part of the ablation zone (20-32 km downglacier distance), increased by 0.3 to 0.6 m.d$^{-1}$, or 50 to 100 %. Overall, the upglacier migration of this acceleration takes about 4 to 5 months. This acceleration is followed by a period of fast velocity lasting from 1 to 3 months over a given area of the glacier. In the following, we refer to this acceleration as the spring/summer acceleration. We observe that the area of fast velocity widens with time, which means that acceleration migrates up the glacier faster than the deceleration does. At the same time, it appears that the average acceleration becomes lower while moving upglacier, with acceleration above 100% on a narrow section near the terminus (> 50 km downglacier distance) in April, and acceleration of the order of 50% in June-July further up the glacier. This period is then followed by slow velocities from August to October, when velocities are about 0.5 m.d$^{-1}$ below the median velocity (range of -50 to -75 %). Between November and December, we observe a short period of fast velocity over the lower part of the glacier (> 40 km downglacier distance) with an acceleration of up to 0.5 m.d$^{-1}$, i.e., 100 to 150 %. This acceleration is well visible in the years 2018 and 2019, and also hinted at in the years 2013 and 2016 (Figure 3a). In the following, we refer to this second acceleration in the ablation zone as the autumn acceleration. In the next section, we show similar velocity patterns from other glaciers in the vicinity.



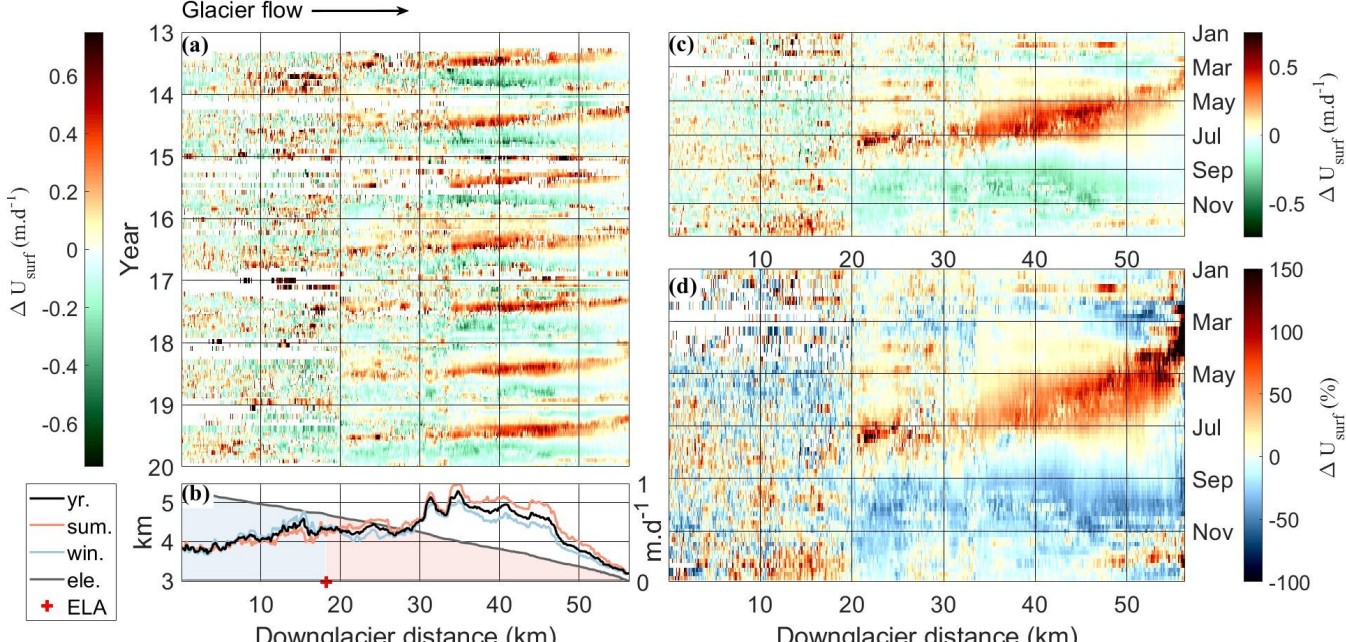

**Figure 3** *(a) Changes in glacier velocity relative to the multi-annual median velocity along the Fedchenko Glacier flowline averaged over 20 days and quantified using Sentinel-2 and Landsat 8 images. The colour bar is coded on a linear scale from green (slower than the median velocity) to red (faster than the median velocity). The black arrow indicates the direction of glacier flow. (b) 7-year*

*median annual (black), summer (March to mid-August; orange) and winter (mid-August to February; green) velocity along the flowline. Light grey line shows the elevation profile, with the blue area located above the ELA and the red area under the ELA. (c, d) Annual profile of absolute (c) and relative (d) changes in glacier velocity relative to the annual median velocity along the Fedchenko Glacier flowline, averaged over 10 days for each year. Changes in % are evaluated relative to the median velocity, i.e. a*

*change of -10% corresponds to a velocity 10% below the median velocity.*

**4.2 Similarities and differences in seasonal glacier velocity**

In Fig. 4, we show the characteristic 10-day annual velocity time series along the flowline of each of the 6 selected glaciers (Figure 1), based on velocity stacking from spring 2013 to winter 2019-





2020. These glaciers cover a range of velocities from rather slow (<0.2 m.d$^{-1}$) to fast (>0.5 m.d$^{-1}$) and of glacier sizes with glacier lengths from less than 5 km to 24 km. However, each velocity time series has its own particularity, which we describe below.

We observe a spring/summer accelerations for all six glaciers. Analogous to the Fedchenko Glacier, velocity changes are larger near the glacier front (>100%) than in the upper part (range of 25 to
75%), except for the MT Glacier (Figure 4j), which shows fast velocities (~150%) along its entire length. The accelerations do not start at the same time for all glaciers, with an onset in early March for the RGS Glacier (Figure 4e, g) and the MT Glacier (Figure 4j, l) and a later onset in May to early June for the T3 Glacier (Figure 4a, c) and the H765Glacier (Figure 4i, k). Furthermore, the accelerations affect the whole length of the RGS (Figure 4e), W731 (Figure 4f), H765 (Figure 4i) and MT (Figure 4j)
glaciers. For the T3 (Figure 4a) and GG (Figure 4b) glaciers, the accelerations are limited to the lower part, as for the Fedchenko Glacier. We note that these three glaciers reach higher altitudes than the four others (See associated elevation profiles in Figures 3c and 4c, d). The upglacier migration of the accelerations is well pictured on 3 of the 6 glaciers (Figure 4a, i, j), which shows that such velocity variations can be observed both on large (e.g., the Fedchenko Glacier) and very small glaciers (e.g.,
H765 and MT glaciers).

After the spring/summer accelerations, there is a period of slow velocity in September followed, in the case of 4 of the 6 glaciers (Figure 4a, e, i, j) by accelerations in autumn. This acceleration is very clear on the MT glacier with velocity changes of the same order of magnitude compared to the spring/summer accelerations. For this glacier, we also observe a clear downglacier migration of the
acceleration. On the other 3 glaciers, this acceleration is of smaller magnitude than the one during spring/summer, and seems to affect only their lower parts. To assess factors controlling the timing and pattern of the periods of faster and slower velocity during the year, we present in the next section the spatio-temporal evolution of the spring/summer and autumn accelerations on the 38 glaciers indicated in Figure 1a (grey lines).










**Figure 4.** *Annual pattern of glacier velocity changes relative to the annual median velocity along the flowline of the selected glaciers in Figure 1a (green lines; except the Fedchenko Glacier). Changes are expressed in % to the annual median velocity. (a, c) Tanymas 3 Glacier; (b, d) Grumm-Grzhimaylo Glacier; (e, g) Russian Geographical Society Glacier; (f, h) Walter 731 Glacier; (i, k) Hadyrsha 765* 370 *Glacier; (j, l)Malyy Tanymas Glacier. The 7-year median annual (black), summer (orange) and winter (green) velocity along the flowline is shown below each glacier. Glaciers flow from left to right (black arrow). Light grey line shows the elevation profile, with the blue area located above the ELA and the red area under the ELA. The colour bar is the same for all glaciers. It is coded on a linear scale from blue (slower than the median velocity) to red (faster than the median velocity). The full sequence of* 375 *glacier surface velocity changes over the period 2013-2020 is available in the supplementary information.*

## 4.3 Characteristics of the spring/summer and autumn accelerations

We detected spring/summer accelerations on 38 glaciers and autumn accelerations on 20 glaciers out of the 48 investigated glaciers (Figures 1, S5).

### 4.3.1 Extraction of the accelerations onsets

For each of these glaciers, we automatically extracted the time and position of the onset of the accelerations along the flowline. To do this, we divided the characteristic annual velocity time series into two periods (red lines in Fig. 5), one between Feb. 15th and Sep. 3rd during which we searched for the onset of the spring/summer accelerations and one between Sep. 3rd and Dec. 12th during which we 385 searched for the onset of the autumn accelerations. We defined these periods according to the velocity patterns previously described (Figure 5). For each period, we created a binary image of the velocity changes by keeping values above 35% for spring/summer and 20% for autumn (Figure S6). In this binary image we remove all components that are connected over a distance less than 1/5th of the glacier length in spring/summer and 1/10th in autumn. We used this image to select the onsets of the 390 accelerations along the flowline (position and elevation; blue dots in Fig. 5). We fitted these automatically picked points with a linear regression to determine the migration rate of the accelerations along the flowline as a function of downglacier distance (blue lines in Fig. 5) and glacier elevation. In





the following, such migration rates are referred to as upglacier (downglacier) migration rate if their earliest onset is located at the lowest (highest) elevation affected by the accelerations. We show in Fig. 6 the onset date of the accelerations as a function of elevation (Figure 6a, c) and distance along the flowline (Figure 6b, d) for the spring/summer and autumn accelerations.

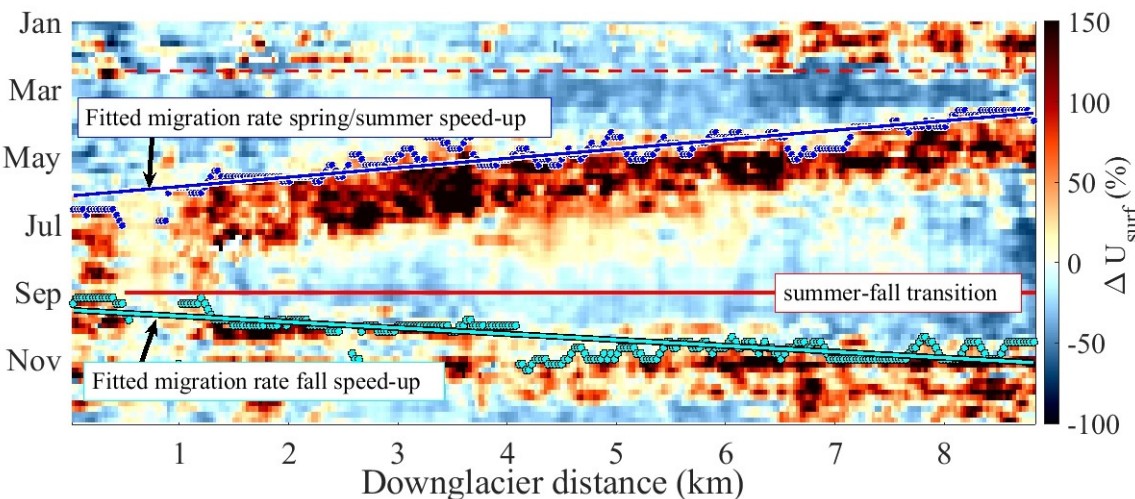

**Figure 5.** *Illustration of our automatically picking of the onset of the spring/summer and autumn accelerations. Background is the annual pattern of relative changes in glacier velocity with respect to the annual median velocity along the flowline of the Malyy Tanymas Glacier (as in Fig. 4j). Dark and light blue dots show automatically picked onset of the spring/summer and autumn accelerations, respectively. Dark and light blue lines show the linearly fitted migration of accelerations as a function of along flow distance. Dashed and solid red lines show the initiation of the spring/summer and autumn periods, respectively.*

### 4.3.2 Temporal characteristics

For the 38 glaciers that exhibit spring/summer accelerations, it usually starts between March and May at the glacier front and between June and August at higher altitudes, illustrating a clear upglacier migration (Figure 6a, b). The rate at which the accelerations migrate with altitude (Figure 6 a) is similar between glaciers (12 ± 9 m.d$^{-1}$), and comparable to the rate at which the mean daily air temperature





410    increases with altitude in spring (18 m.d$^{-1}$; black line in Fig. 6a). The onset of accelerations occurs about one month before the mean daily air temperature becomes positive (black line in Fig. 6a).

For the 24 glaciers that exhibit an autumn acceleration, it usually starts in September in the upper part of the glaciers and a few weeks to a few months later towards the glacier front, illustrating a downglacier migration (Figure 6c, d). The rate at which this downglacier migration occurs (34 ± 18 m.d$^{-1}$) is twice as fast as the rate of upglacier migration during spring/summer. However, these rates are still comparable to the rate at which the mean daily air temperature decreases with altitude in autumn (24 m.d$^{-1}$; black line in Fig. 6c). The onset of acceleration occurs within one month after the mean daily air temperature becomes negative (black line in Fig. 6c). Furthermore,  we do not observe a significantly different geometry (i.e., slope, width, length, orientation, hypsometry) or dynamics between glaciers that exhibit or not such acceleration (Figure S5).



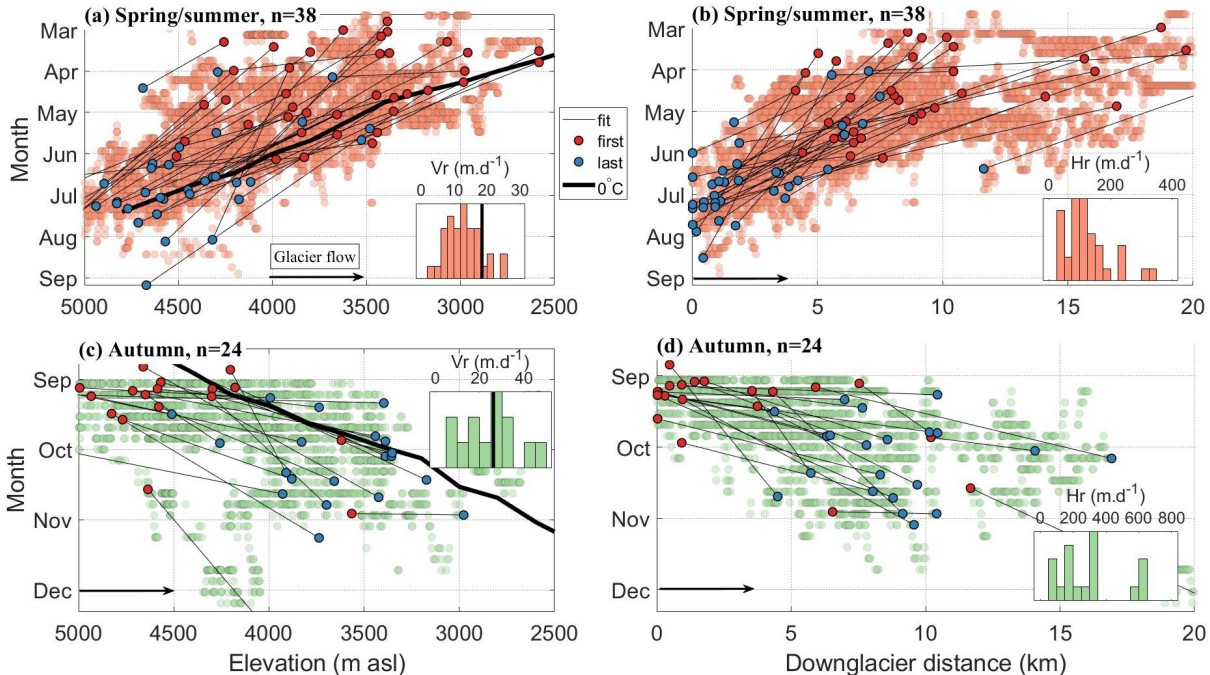

**Figure 6.** *Observed time at which the accelerations starts along the flowline as a function of (a, c) the associated glacier elevation and (b, d) the downglacier distance for the spring/summer period (top panels) and the autumn period (bottom panels). Light orange and green dots show the automatically picked points (as described in Fig. 5), red points show the earliest onset date of the accelerations for each glacier and blue points the latest onset date. Gray lines show the fitted migration rate of the accelerations along the flowline for each glaciers. Histograms show the distribution of the vertical (Vr) and horizontal (Hr) migration rate. Bold black line in (a) shows the time and elevation at which daily averaged air temperature starts to be positive during the melt-season; above this line temperature are negative and under this line they are positive (Figure S1). Bold black line in (c) the time and elevation at which air temperature drop to negative values at the end of the melt-season; above this line temperature are positive and under this line they are negative (Figure S1). Migration rates of these temperature changes with altitude are shown in bold black lines in the histograms.*

## 5 Discussion

### 5.1 Variations in migration rate





The onset of the spring/summer accelerations appears to be positively correlated to the elevation at which it occurs (Figure 6a, b): the accelerations start earlier at lower altitudes and end later at higher altitudes (Figure 7a, b). These trends do not appear to be influenced by the glacier orientation (marker color) or slope (marker size); instead, they appear to be similar to the trend in temperature change with altitude (plain lines in Fig. 7a, b). The accelerations generally start in the two months before positive daily mean air temperature are reached at the beginning of the melt-season (Figure 6a, b; Figure 7a, $R^2$ of 0.17), and they reach their highest elevations when positive daily maximum air temperatures are reached at the end of the melt-season (Figure 6a, b; Figure 7b, $R^2$ of 0.33). For some glaciers, the fraction of the glacier subject to acceleration appears to be positively related to the fraction of the glacier below the ELA (Figure 7c). Glaciers that accelerate over 80 to 100 % of their extent, however, defy this trend. The upglacier migration rate of the accelerations (i.e., the norm of the vertical and horizontal rates) appears to be negatively related to the slope of the glacier (Figure 7d). In other words, the shallower the glacier slope, the faster the migration rate. When normalizing the migration rate by the glacier slope (Figure 7d, c), we observe no obvious influence of the median glacier velocity on the migration rate (Figure 7e) but a positive relation with the glacier width (Figure 7f, $R^2$ of 0.63). However, when distinguishing by normalized migration rate, for glaciers with a higher rate (> 1 km.d$^{-1}$), the migration rate tends to decrease with increasing glacier velocity.

Similarly to the spring/summer accelerations, the timing of the autumn accelerations is related to the levation at which it occurs (Figure 6c, d): the accelerations start earlier at higher altitudes and end later at lower altitudes, thus depicting a clear downglacier migration. The fraction of the glacier subject to accelerations does not seem to depend significantly on the fraction of the glacier below the ELA (Figure 8c). Most of the onset dates (~80%) occur after the daily mean air temperature has reached negative values at the end of the melt-season (Figure 8a, b). We observe that the autumn accelerations are restricted to a smaller and lower area (i.e., smaller fraction of the glacier; Figure 8c) than during the spring/summer accelerations, which is also indicated in Fig. 4. These patterns are not clearly influenced by the glacier slope (marker size) or the glacier orientation (marker color). As during the spring/summer accelerations, the migration rate of glacier acceleration appears to be negatively related to the glacier slope (Figure 8d). When normalizing the migration rate with the slope of the glacier, there is no clear





465    influence of the glacier velocity on the downglacier migration rate. However, as for the upglacier

migration rates, we observe that the downglacier migration rate increases with glacier width (Figure 8f,

$R^2$ of 0.68).



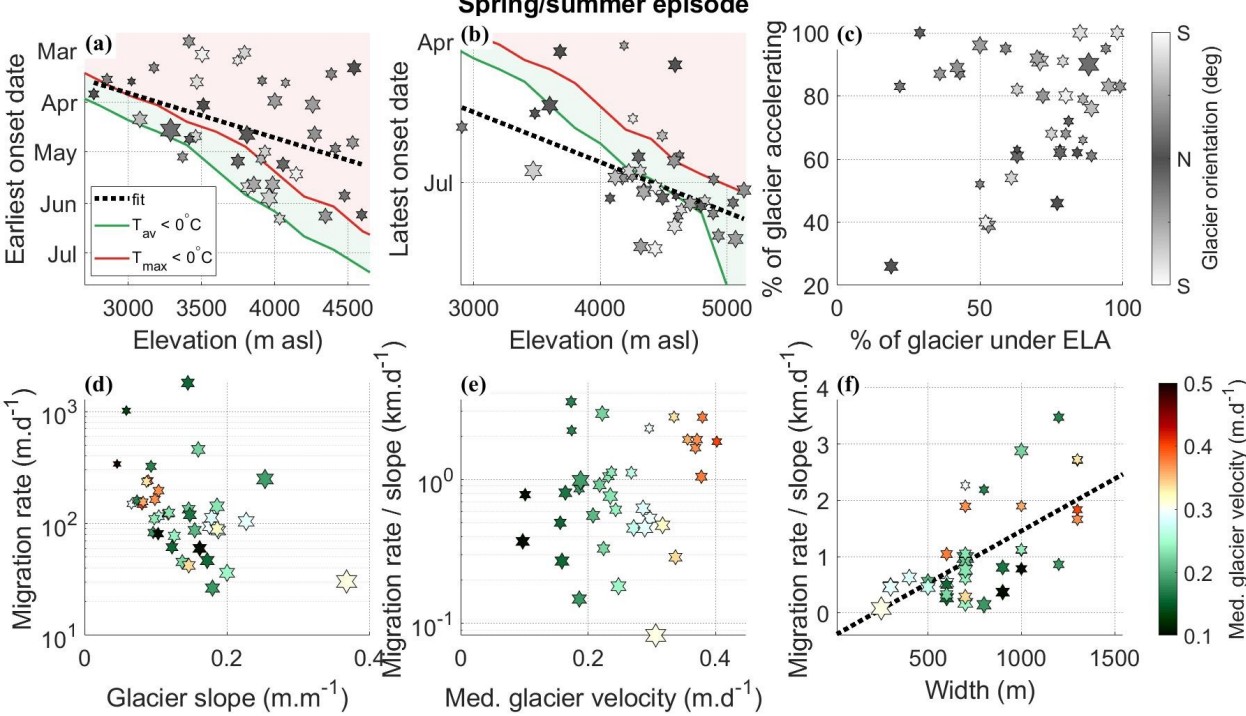

**Figure 7.** *Characteristics of the upglacier migration rate of the spring/summer accelerations measured on 38 glaciers. (a) Earliest and (b) latest onset dates of the accelerations as a function of the glacier altitude where they occur. The dotted lines show a linear fit with $R^2$ of 0.17 and 0.33 in (a) and (b), respectively. The red and green lines in (a, b) show when the daily maximum and daily mean temperature start to be positive during the melt-season (Figure S1). (c) Fraction of glacier length over which the accelerations are observed, relative to the fraction of glacier length below the ELA. Marker colors in (a-c) show glacier orientation. (d) Migration rate as a function of mean glacier slope. (e, f) Migration rate normalized by mean glacier slope as a function of median glacier velocity and glacier width, respectively. The dotted line shows a linear fit with $R^2$ of 0.63. Marker colors in (d-f) show median value of the velocity reached during the accelerations. The size of the marker represents the slope of the glacier, the smaller the marker the shallower the slope.*




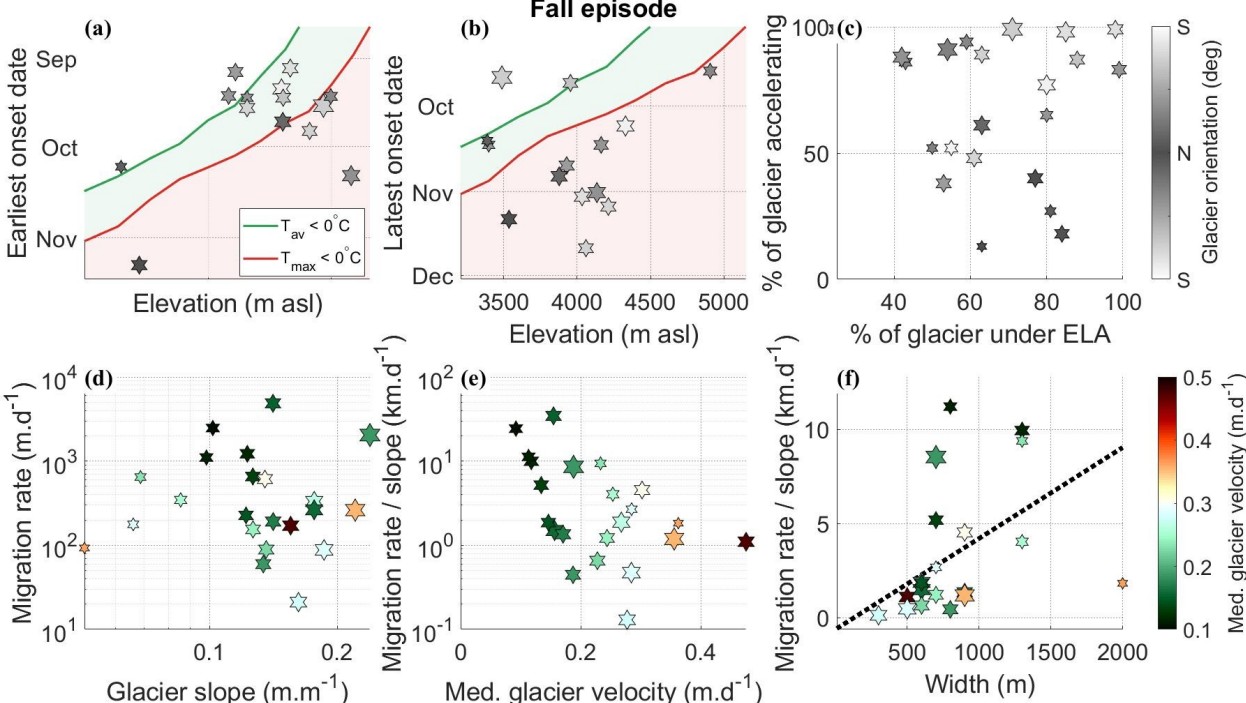

**Figure 8.** *Characteristics of the downglacier migration rate of the autumn accelerations measured over 24 glaciers. (a) Earliest and (b) latest onset dates of accelerations as a function of glacier elevation*

485 *where they occur. The red and green lines in (a, b) show when the daily maximum and daily average start to be negative at the end of the melt-season. (c) Fraction of glacier length over which accelerations are observed relative to the fraction of glacier length below the ELA. Marker colors in (a-c) show glacier orientation. (d) Migration rate as a function of mean glacier slope. (e, f) Migration rate normalized by mean glacier slope as a function of median glacier velocity and glacier width,*

490 *respectively. The dotted line shows a linear fit with $R^2$ of 0.68. Marker colors in (d-f) show median value of the velocity reached during the accelerations. The size of the marker represents the slope of the glacier, the smaller the marker the shallower the glacier.*





### 5.2 Causes of short-term velocity changes

**5.2.1 Spring/summer**

Of the 48 glaciers we studied, 38 of them show clear changes in glacier velocity between a slow winter period (November to February), and a fast spring/summer period (March to August) when many glaciers accelerate by up to 150-300% (Figurs 3, 4). This behaviour has already been observed on the Fedchenko Glacier (Lambrecht et al., 2014) and other large and/or fast-flowing mountain glaciers, for instance in the neighboring Karakoram range (Quincey et al., 2009; Scherler and Strecker, 2012; Usman and Furuya, 2018), in the European Alps (Gordon et al., 1998; Vincent and Moreau, 2016), in the Alps of New Zealand (Purdie et al, 2008) or in Alaska (Armstrong et al., 2016), but has not yet been documented on such a variety of mountain glaciers with high temporal resolution (Figures 1, 4).

Our results show that the spring/summer accelerations are mainly confined to the ablation zone and depicts an upglacier migration, which appears mainly controlled by altitude and air temperature (Figures 4, 8). Elevation and air temperature are linearly related and exert a key control on melt rates, which is the main contributor to bed water supply in spring/summer (Yao et al., 2012; Lambretch et al., 2018). We suggest that the spring/summer accelerations are primarily caused by the evolution of the subglacial drainage system in response to changes in bed water supply. At the beginning of the melt-season (March to May) and at low elevations (~3000 m) the supply of meltwater to the bed increases rapidly due to positive air temperatures (Figure S1) and inundates a likely inefficient drainage system, resulting in increased basal pressure and consequently higher glacier velocity (Lliboutry, 1968; Iken and Bindschalder, 1986). This is supported by the general onset of accelerations at the lowest elevations (Figure 6a, b, Figure 7a). Later during the melt-season, positive temperatures are reached at higher altitude (Figure 6) and upper parts of the ablation zone are subjected to increased supply of meltwater to the bed, which likely induces an increase in basal slip. This is supported by the upglacier migration of the accelerations (Figure 6) and the later onset of the accelerations at high elevation (Figure 7a). Concomitantly, meltwater supply to the bed continues in the lower parts of the ablation zone, likely leading to the development of an efficient drainage system, which decreases the basal water pressure and thus the glacier velocity (Lliboutry, 1968; Iken and Bindschalder, 1986; Nanni et al., 2021). Indeed,



a slowdown following an acceleration is observed towards the glacier front between May and July (Figures 3, 4).

The spring/summer accelerations appear to be also influenced, although to a lesser extent, by glacier dynamics and geometry, as the rate of upglacier migration significantly increases with width and slightly decreases with glacier median velocity (Figure 7d, e, f). Very fast migration rates (corrected by the influence of slope) mean that the accelerations occur almost at the same time along the entire length of the glacier. This kind of behaviour could be promoted by glaciers where the drainage system responds quickly to water supply to the bed and/or the drainage system is very inefficient at the beginning of the melt-season. Such conditions are expected for thick glaciers, which are typically also wide glaciers, as higher normal stress favors closure of the efficient drainage system during the winter period (Llibourtry, 1968) and for slow moving glaciers where the drainage system is expected to be less efficient than for fast moving glaciers (Kamb, 1987).

### 5.2.2 Autumn

20 out of the 48 glaciers we analyzed show clear accelerations in autumn, followed by a period of faster velocities (Figures 3, 4, 6). Such autumn accelerations has rarely been observed before and we discuss here its potential cause(s). As during spring/summer, the accelerations do not occur at the same time over the glacier. It often describes a downglacier migration related to the evolution of air temperature with elevation (Figure 6c, d). The period of faster velocity subsequent to the autumn accelerations is of shorter duration (~1 month) than the one following the spring/summer accelerations (~1-4 months) and these faster velocities are often restricted to the lower part of the ablation zone (Figures 6, 8).

Acceleration periods during autumn of shorter duration (e.g., from days to weeks) have been observed previously and have been proposed to be caused by a reduced efficiency of the drainage system making it more sensitive to sudden water supply (Hodge, 1974; Sugiyama and Gudmundsson, 2003; Harper et al., 2005; Hart et al., 2019). Gradual closure of the drainage system occurs when the water supply to the bed decreases and does not counteract the creep closure (Röthlisberger, 1972), which is probably the case for mountain glaciers at the end of the melt-season (Nanni et al., 2020). As a





consequence, the efficiency of the drainage system decreases, making it more sensitive to low water
input. This process is consistent with the downglacier migration that we observe, as closure should first
occur higher up, where meltwater input decreases earlier. It is also consistent with the fact that we
observe such episodes mostly in the lower part of the glacier, as enough water is needed for the drainage
system to be pressurized again. This water input could be due to drainage from supraglacial ponds
(Clason et al., 2015; Miles et al., 2018), rainfall events (Vieli et al., 2004; Horgan et al., 2015) or
gradual release of subglacially-stored water, which has been observed to promote sustained high
velocity outside the melt-season (Tedstone et al., 2013; Hart et al., 2019). In the Western Pamir, the
autumn season is typically dry (Yao et al., 2012), making it unlikely that rainfall events are the main
cause for the water input. Of the glaciers that exhibit this autumn accelerations, around 50% have an
ablation zone with significant debris-cover (Mölg et al., 2018) and most of them show the presence of
supraglacial ponds in the upper ablation zone (Figure S7 in supporting information). Such ponds are
generally small (<100 m wide) and are similar to those observed in the Himalayan regions where the ice
is crevassed and/or debris covered (Iwata et al., 1980; Sakai et al., 2000; Miles et al., 2017).
Supraglacial ponds tend to form during the summer period and drain in late summer and/or early
autumn (Miles et al. 2018), which is consistent with the timing of the autumn accelerations. We
hypothesize that these drainage events could contribute to the sudden influx of meltwater to the bed in
autumn. Water storage in firn has also been observed to delay runoff from days to months (Jansson et
al., 2003); however very few studies have been conducted in the Pamir region, which makes it difficult
for us to assess the extent to which water storage may occur.

We therefore suggest that the autumnal accelerations result from glacier instability occurring in
the upper ablation zone and propagating down the glacier. This instability could be caused by a sudden
influx of meltwater, as discussed above, which triggers a local increase in basal water pressure,
facilitated by a reduced capacity of the drainage system. This leads locally to a rapid change in basal
stress, which can promote the propagation of an instability in a manner similar to surges (Thøgersen et
al., 2019, Beaud et al., 2022) or glacier response to calving (Riel et al., 2021). The downward
propagation reaches migration rates similar to those observed on alpine glaciers in the form of a
kinematic wave (Hewitt and Fowler, 2008), which supports our hypothesis of a local disturbance. Such





sensitivity to local disturbance may be favored by soft-bedded glaciers, which is probably the case for the glaciers studied, since 70% of them are sugggested to be surge-type (Goerlich et al., 2020). If our hypothesis that meltwater input by supraglacial pond drainage holds true, these events could represent

the lower end of hydraulically controlled surges and glacial lake outburst floods (GLOFs), which play an important role in mountain natural hazards (Iribarren et al., 2018; Kääb et al., 2018; Bhambri et al., 2019; Yang et al., 2022).

### 5.3 Limits and opportunities in monitoring short-term velocity changes

Our approach resolves peak velocity changes as low as $\pm0.1$ m.d$^{-1}$ (36.5 m.year$^{-1}$), for glacier
with a median velocity as low 0.1 m.d$^{-1}$ (Figure 1, 4). Where the median velocity is of the order of 0.2 m.d$^{-1}$ (73 m.year$^{-1}$), the observed accelerations can be as low as 25% (-0.05 m.d$^{-1}$, -18.25 m.year$^{-1}$). We observe such changes for large glaciers (e.g., Fedchenko Glacier), but more importantly also for small glaciers that are less than 5 km long and 250 m wide (Figure 4e). With pixel sizes of 10 and 15 m, for Sentinel 2 and Landsat 8, respectively, this corresponds to a precision of 1/10$^{th}$ to 1/15$^{th}$ of the pixel for
low velocity areas ($\sim$0.1 m.d$^{-1}$) and of 1/20$^{th}$ to 1/30$^{th}$ of the pixel for mid-to-fast velocity areas (>0.2 m.d$^{-1}$). Such a sub-pixel image matching precision is higher than the 0.15 to 0.33 pixels that we calculated based on stable ground analysis (Figure 2). It is also higher than the 0.1-pixel precision previously estimated for mountain areas (Scherler et al., 2008; Heid and Kääb, 2012). We suggest that the high precision we obtain when investigating peak velocity is due to two reasons. On the one hand,
the stable-ground estimation might overestimate the error, especially in our study area which is mainly covered by mountainous areas where the standard deviation over ice-free areas can also be affected by natural surface displacement such as hillslope movements or riverbed changes. The precision obtained over glaciated area, where most of the displacements are caused by glacier flow, should not be influenced by displacements observed over ice-free areas and could therefore be higher (more precise).
In addition, glaciers have typically lower slopes than their surroundings, which reduces the impact of inaccuracies in the elevation model used for the orthorectification (Scherler et al., 2008). Indeed, we observe significant changes in surface velocity of less than 0.1 m.d$^{-1}$ over the shallow-sloping ablation zone over 10 to 20-day periods (Figure 4), which support a matching precision on the order of 0.1 pixel

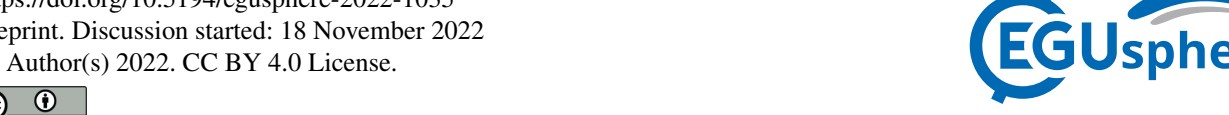

in such areas. This higher precision could also be due to faster flowing glacier and more textured
surfaces in the ablation zone compared to the accumulation zone and stable ground (Figures S8, S9). On
the other hand, we suggest that by combining multiple measurements (different sensors, time periods
and years), we obtain an improved signal of seasonal and recurring glacier velocity changes. Such an
approach has been shown to significantly increase measurement precision (Altena et al., 2019;
Derkacheva et al., 2020). We have observed that Sentinel-2 imagery can help to resolve patterns that are
only hinted at using Landsat-8 only. However, the increase in spatial and temporal resolution obtained
with the Sentinel-2 imagery does not offer a significant advantage as shown by the similar resolution we
obtain (Figure 2). We suggest that further improvement of our protocol could be made using the
approaches presented in Altena et al. (2019) and Derkacheva et al. (2020), which would allow
combining velocity obtained from different sensors and over different time spans with more advanced
statistical approaches.

The high precision we obtain shows the advantage of combining a statistical approach with a
robust measurement of surface displacement to improve the level of comprehension on large satellite
imagery datasets. This is made possible by the COSI-Corr algorithm and the medium-resolution high
quality satellite imagery we have used. This approach takes advantage of the large datasets that are now
available in remote sensing and thus allows monitoring short-term (down to 10-day) changes in
mountain glacier velocity, even for relatively slow flowing glaciers (0.1 m.d$^{-1}$) and/or on the slower
parts of glaciers.

**6 Conclusions**

Our study demonstrates that the large amount of optical imagery that is now available from
satellite programs such as Landsat-8 and Sentinel 2 can be effectively exploited to monitor surface
velocity of mountain glaciers with a resolution down to 10 days. Our protocol requires minimal manual
processing and is effective for a wide range of glacier geometries, from small (~250 m wide, 5 km long)
to large (> 50 km long). This protocol is not only applicable to glaciers but also to other types of surface

movements. We observed a clear pattern of seasonal velocity changes that repeats over 7 years on 38 glaciers in the Western Pamir, with accelerations in spring-summer that start at the glacier front and propagate upwards. The upglacier migration rate correlates with the air warming rate, which supports a climatic driver. We link these accelerations to changes in the subglacial drainage system in response to changes in meltwater supply to the bed. We also observed autumn acceleration episodes on 24 glaciers

that occur on an annual basis in the ablation zone and for which the accelerations propagate downglacier in relation with the air cooling rate,. We link these episodes to local glacier instabilities, probably caused by drainage of supraglacial ponds and changes in drainage system efficiency. This shows the potential of our approach to highlight not only the well-known seasonal accelerations, but also shorter-term dynamics, which are crucial to capture in order to study the response of glaciers to

changes in air temperature.

**Code and data availability**

The glacier surface velocity maps as well as the along flowline velocity profiles data used for in the study are available at https://doi.org/10.5281/zenodo.7142273. (Nanni, 2022b) under a Creative Commons Attribution 4.0 International license.

V.1.0 of the codes we developed to conduct our study (as detailed in Section 3) is preserved at https://doi.org/10.5281/zenodo.7142273 (Nanni, 2022b) under a Creative Commons Attribution 4.0 International license. The codes are to be used in combination with ENVI's COSI-Corr add-on software, which is freely available at http://www.tectonics.caltech.edu/slip_history/spot_coseis/ (last access: September 9[th] 2022; Leprince et al., 2007). An python version of COSI-Corr is currently under

development and can be found here: https://github.com/SaifAati (last access: October 5[th] 2022).

**Video Supplement**

Seasonal variations in glacier surface velocity over the Western Pamir can be visualized here: https://imgur.com/a/Ugdh4UJ (last access: October 4[th] 2022).



**Author contribution**

Conceptualization: UN, DS, FH, J-PA; Data curation: UN, FA, DS; Investigation: UN, DS; Methodology: UN, DS, FH, RM; Software: UN, FA, DS; Writing: UN, DS, RM, FH.

**Competing interests**

The authors declare that they have no conflict of interest.

**Acknowledgments**

U.N thanks B. Altena, F. Brun, A. Gilbert, A. Kääb and L. Malatesta for helpful comments and discussions. U.N greatly thanks J. Mouginot for helping with the processing and the discussions. The authors thank A. Lambretch for providing meteorological data on the Fedchenko Glacier (Figure S1). D.S acknowledges funding from the European Research Council under the European Union's Horizon 2020 research and innovation programme under grant agreement 759639. This research was partially

funded by the Resnick Sustainability Institute at the California Intsitute of Technology.

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
