# Peer review of "Climatic control on seasonal variations of glacier surface velocity"

_EGUsphere, 2022_

## Referee Comment (RC1)

**Review of Nanni et al., The Cryosphere – Peter Tuckett (University of Sheffield)**

**General Comments**

This paper presents high temporal resolution time series of ice velocity from 48 mountain glaciers in the Western Pamir. The manuscript provides a detailed description of the feature tracking method, and successfully demonstrates how ice velocity measurements can be generated from optical satellite data in mountainous regions. Although the main method (COSI-Corr) is not novel, the stated filtering and post-processing steps advance the applicability of the method in mountain settings. Multi-year, weekly/monthly time series of satellite-derived glacier velocities in mountain regions are relatively sparse, and this dataset is therefore a useful contribution.

The most interesting scientific result is the identification and discussion of autumn speed-up events. These are less well documented than spring events, and I was interested by the discussion of their cause. I agree that these autumn speed-ups are likely the result of supraglacial input to an inefficient drainage system, but think this section would benefit from some expansion. In particular, I think there is further evidence in the velocity data for a subglacial hydrological cause which has been missed, and the section would benefit from accompanying evidence of lake drainage events from optical imagery. These would be relatively quick changes/additions, but could help to provide further evidence for the suggested cause.

I would also like to see the authors provide more of a discussion of the limitations of velocity measurements from optical satellite imagery, and how these were overcome. In particular, there is very limited explanation of how spatially and temporal variable cloud cover is treated in the method. It is stated that the region has low cloud-cover, and that cloud-pixels are 'flagged', but it is not really explained how differences in image visibility are accounted for to generate continuous and spatially consistent velocity time series.

The work is scientifically robust and presents some interesting results. The manuscript is generally well presented, although I think some of the figures could be presented more clearly, especially the colours used. The text is well written and clearly structured, although a few points need some clarification (see specific comments). The methods are well explained, and I particularly like the overall effect of figures 3-5. I would be happy to see this work published after minor edits based on the following comments and suggestions.

**Specific Comments**

Title: I would suggest that the title is unnecessarily vague, and should at least specify that the work is related to a mountain setting, if not the region itself.

Line 94: Specify the stated recurrence times are specific for this region, since this varies spatially (e.g. there are much shorter return periods in polar regions).

Line 115: 'Different characteristics and geometries' is very vague and could be applicable to any mountainous regions – be more specific on why this region was selected.

Line 132: This wording confused me slightly – what did you investigate if you didn't analyse the data? Do you mean that velocity data were generated for 48 glaciers, but only 38 showed seasonal variations? Suggest clarifying.

Line 133: Glacier geometries? 'Wide range of glaciers' is ambiguous.

Figure 1: I found this figure a bit messy. E.g. (c) has a black outline for some of the inset but not all, GG label goes outside the main box, (b) label is not aligned with (a) label etc. Colours showing background elevation in (b) do not add anything and does not have a colourbar, so I suggest simply removing and having a white background.

Line 153: Just state the years rather than 'time periods above'.

Line 163: Why does Sentinel-2 data only start in 2017 when the satellite was launched in 2015?

Line 187: It states that you 'flag' clouds pixels, but there is then no explanation of what you do with these pixels. Even in a low-cloud region, a critical limitation of optical derived velocity data is that cloud cover can result in incomplete or inconsistent time series. This hence requires some further explanation.

Line 255: This last sentence seems a bit out of place and unnecessary. Suggest removing.

Line 260: Were central flowlines manually drawn?

Line 306: See comment for Figure 3. You mainly refer to distances upglacier, so it then seems confusing to have to specifiy downglacier values in brackets simply to match the figure.

Figure 3: Instinctively, I think upglacier distance would be better for the x-axis, with the axis values flipped. There are also a lot of similar colours in (b) which I think could be made clearer. Both these comments also apply for Figures 4 and 5.

Figure 5: The text boxes block some of the interesting velocity data, in particular relating to the autumn speed-up events between 1 -3 km downglacier. These are stated in the caption, so I suggest removing from the figure.

Line 538: 'It often describes' – suggest rephrasing.

Line 565: Have the authors looked at the optical imagery to see if lake drainage events are visibile? It might not show anything (it could be englacial drainage which is harder to observe visually), but if it did, it would help to validate the suggested mechanism. Figure S7 shows lakes identified in the region, but it would also be useful to present some close-up images of these lakes to give an indication of their size.

Line 569: Figure 5 seems to show a sudden, transient autumn speed-up immediately followed by a slow-down to below pre-acceleration velocity. This is particularly clear at higher elevations (shown by a thin red stripe, then a thin blue stripe between 1 – 4 km downglacier). This is consistent with what would be expected from the sudden input of water to an inefficient subglacial hydrological system, but is not mentioned.

Line 583: As previously mentioned, I think this section should include something on the limitations of optical data in general for velocity retrieval.

**Technical corrections**

Line 140: Black/white colours don't match figure

Line 243: Don't think you mean Figure 1?

Line 455: 'levation' to 'elevation'

Line 513: Specify 'basal water pressure'

---

## Author Comment (AC1)

**Peter Tuckett**

**General Comments**

This paper presents high temporal resolution time series of ice velocity from 48 mountain glaciers in the Western Pamir. The manuscript provides a detailed description of the feature tracking method, and successfully demonstrates how ice velocity measurements can be generated from optical satellite data in mountainous regions. Although the main method (COSI-Corr) is not novel, the stated filtering and post-processing steps advance the applicability of the method in mountain settings. Multi-year, weekly/monthly time series of satellite-derived glacier velocities in mountain regions are relatively sparse, and this dataset is therefore a useful contribution.

The most interesting scientific result is the identification and discussion of autumn speed-up events. These are less well documented than spring events, and I was interested by the discussion of their cause. I agree that these autumn speed-ups are likely the result of supraglacial input to an inefficient drainage system, but think this section would benefit from some expansion. In particular, I think there is further evidence in the velocity data for a subglacial hydrological cause which has been missed, and the section would benefit from accompanying evidence of lake drainage events from optical imagery. These would be relatively quick changes/additions, but could help to provide further evidence for the suggested cause.

I would also like to see the authors provide more of a discussion of the limitations of velocity measurements from optical satellite imagery, and how these were overcome. In particular, there is very limited explanation of how spatially and temporal variable cloud cover is treated in the method. It is stated that the region has low cloud-cover, and that cloud-pixels are 'flagged', but it is not really explained how differences in image visibility are accounted for to generate continuous and spatially consistent velocity time series.

The work is scientifically robust and presents some interesting results. The manuscript is generally well presented, although I think some of the figures could be presented more clearly, especially the colours used. The text is well written and clearly structured, although a few points need some clarification (see specific comments). The methods are well explained, and I particularly like the overall effect of figures 3-5. I would be happy to see this work published after minor edits based on the following comments and suggestions.

We sincerely thank the reviewer for such positive and clear summary of our work, as well as for his thoughtful comments that we carefully address in the following. We appreciate the above suggestions which have helped improve the manuscript.

**Specific comments of RC1:**

Title: I would suggest that the title is unnecessarily vague, and should at least specify that the work is related to a mountain setting, if not the region itself.
We specified that the work is related to mountain glaciers.

Line 94: Specify the stated recurrence times are specific for this region, since this varies spatially (e.g. there are much shorter return periods in polar regions).
This was changed accordingly in the main text.

Line 115: 'Different characteristics and geometries' is very vague and could be applicable to any

mountainous regions – be more specific on why this region was selected.

We selected this region as it hosts a wide range of glacier, i.e., very small and very large, and also because of its very low-cloud cover (which is not the case for the Alaskan range or the Karakorum). It now reads:

We then apply our methodology in the Western Pamir region to investigate its performance, as this region features glaciers with different characteristics (e.g., velocities) and geometries (e.g., glacier length) as well as low cloud cover, compared to other regions such as the Karakorum or the Alaskan range

Line 132: This wording confused me slightly – what did you investigate if you didn't analyse the data? Do you mean that velocity data were generated for 48 glaciers, but only 38 showed seasonal variations? Suggest clarifying.

This was clarified. It now reads:

*Our study is focused on a 60 km x 60 km area that is approximately centered on the Fedchenko Glacier. We selected 38 out of 48 studied glaciers for further analysis based on the presence of clear seasonal variations and limited surge occurrence (lines in Fig. 1c)*

Line 133: Glacier geometries? 'Wide range of glaciers' is ambiguous.

This was changed accordingly in the main text.

It now reads: This selection contains a wide range of glacier geometries (see Fig. 1 caption for abbreviations)

Figure 1: I found this figure a bit messy. E.g. (c) has a black outline for some of the inset but not all, GG label goes outside the main box, (b) label is not aligned with (a) label etc. Colours showing background elevation in (b) do not add anything and does not have a colourbar, so I suggest simply removing and having a white background.

We have adjusted the small details shown above. We have kept the coloured background of panel (b) for aesthetic reasons, and it is not necessary to add a colour bar.

Line 153: Just state the years rather than 'time periods above'.

This was changed accordingly in the main text. It now reads:

We downloaded Landsat 8 and Sentinel 2 images that cover our study area for the time period 2013-2020

Line 163: Why does Sentinel-2 data only start in 2017 when the satellite was launched in 2015?

Because it's availability in this area started in 2017, we have added this precision in the text.

Line 187: It states that you 'flag' clouds pixels, but there is then no explanation of what you do with these pixels. Even in a low-cloud region, a critical limitation of optical derived velocity data is that cloud cover can result in incomplete or inconsistent time series. This hence requires some further explanation.

We added:

*We used the quality assessment band for Landsat 8 images to flag and remove pixels where clouds*

Line 255: This last sentence seems a bit out of place and unnecessary. Suggest removing.
The sentence was removed.

Line 260: Were central flowlines manually drawn?
Yes, and this is now stated explicitly in the main text.

*We then calculated the median velocity over the entire time period at each pixel location and used the resulting map (Figure 1) to define the central glacier flowline (here done manually).*

Line 306: See comment for Figure 3. You mainly refer to distances upglacier, so it then seems confusing to have to specifiy downglacier values in brackets simply to match the figure.
Figure 3: Instinctively, I think upglacier distance would be better for the x-axis, with the axis values flipped.

We thank the reviewer for pointing this out. We have modified to upglacier distance throughout the text and the figures.

There are also a lot of similar colours in (b) which I think could be made clearer. Both these comments also apply for Figures 4 and 5.

We have kept the colour code in (b) and for Figs. 4 and 5 because it consists of only 3 colours. The colours used to define the ablation and accumulation zones are the same as those used to define the summer and winter velocity. We now use  more bright colours. The thickness of the lines is sufficient to make it clear that the three curves represent glacier velocity while the shaded areas represent glacier areas and glacier elevation. We have made the line thicker and dashed the line for the elevation to make it clearer.

Figure 5: The text boxes block some of the interesting velocity data, in particular relating to the autumn speed-up events between 1 -3 km downglacier. These are stated in the caption, so I suggest removing from the figure.

We have modified the figure to remove the text boxes.

Line 538: 'It often describes' – suggest rephrasing.
We modified to 'It often follows'

Line 565: Have the authors looked at the optical imagery to see if lake drainage events are visibile? It might not show anything (it could be englacial drainage which is harder to observe visually), but if it did, it would help to validate the suggested mechanism. Figure S7 shows lakes identified in the region, but it would also be useful to present some close-up images of these lakes to give an indication of their size.

Thanks, that's a great suggestion. As noted by the reviewer, we show in Figure S7 a map of the likely location of supraglacial lakes/ponds. Given the very small size of the lakes (< 100 m) and the fact that they are often located in crevassed areas, we have not observed many clear drainage events. We have added a close-up image of a potential lake drainage in the supplementary material to further illustrate this mechanism.

Line 569: Figure 5 seems to show a sudden, transient autumn speed-up immediately followed by a slow-down to below pre-acceleration velocity. This is particularly clear at higher elevations (shown by a thin red stripe, then a thin blue stripe between 1 – 4 km downglacier). This is consistent with what would be expected from the sudden input of water to an inefficient subglacial hydrological system, but is not mentioned.

We thank the reviewer for pointing out this event and we have mentioned it in the text.

*Such a sudden change can be seen in Figure 5, with a rapid increase in velocity directly followed by a drop.*

Line 583: As previously mentioned, I think this section should include something on the limitations of optical data in general for velocity retrieval.

We have added:

*The incorporation of radar images in combination with optical images (Derkacheva et al., 2020) would also increase the accuracy of the velocity field as radar images can be used even for periods of time with cloud cover.*

Technical corrections
Line 140: Black/white colours don't match figure
This was changed accordingly in the main text.

Line 243: Don't think you mean Figure 1?
We do not understand why it should be Figure 1. Figure 1 is the map and Figure 2 the standard deviation analysis. We refer to Figure 1 to show the difference between glaciated and non glaciated areas.

Line 455: 'levation' to 'elevation'
This was changed accordingly in the main text.

Line 513: Specify 'basal water pressure'
This was changed accordingly in the main text.

---

## Author Comment (AC2)

**General Comment**

Nanni et al. developed a processing protocol for the detection and assessment of short-term glacier velocity changes from satellite EO and applied the method in an interesting study on seasonal variations of glacier velocity in the Pamir mountains (2013-2020) and its connections with local climate. The primary source data for generating high spatial- and temporal-resolution time series of ice velocity are NASA/USGS Landsat 8 and Copernicus Sentinel-2 optical satellite imagery, which are processed using the open source COSI-corr software. The post-processing steps include an extensive filtering procedure, calibration and central flowline velocity extraction. In this way the authors analyzed tens of glaciers with different characteristics and generated detailed spatiotemporal velocity plots for investigating seasonal variations in ice velocity.

Most of the glaciers exhibited an annual recurring speed-up in the spring/summer and roughly half a lesser pronounced additional speed-up in autumn/winter, separated by slower velocities in between. Looking in more detail the spatiotemporal pattern revealed an upstream migration over time for the spring/summer speed up and a downstream migration for the autumn/winter speed up in most cases. Using the long-term averaged (1933-1995) air temperature dataset from a nearby station the authors find a good correlation of the evolution (timing and location/elevation) of the speed-up events with the migration of the isotherms, while other factors seem to have lesser pronounced or no relation (slope, orientation, velocity). The authors further provide a convincing discussion on the role of the efficiency of the subglacial hydrologic system, varying during the year, as a main driver for the observed accelerations.

The topic of this paper, seasonal variations of glacier surface velocity and its climatological controls, is very interesting and relevant, in particular thanks also to recent advances in modern computing technology and increasing availability of satellite EO data. This paper by Nanni et al. is a well written, illustrated and referenced manuscript and a valuable and original contribution of interest for the glaciology and wider community. The authors givea good motivation for their work, a detailed description of their methods and results and provide a set of interesting observation and a thorough discussion. The outcome provides new insights on seasonal variability of glaciers and environmental drivers in particular relevant for glacier and climate research. Some specific comments, corrections and suggestions for improvements are provided below.

*We sincerely thank the anonymous referee for providing such a positive summary of our work. We have addressed the different comments and suggestions, and this has helped improve the manuscript.*

**Specific comments:**

Pg 2 – Ln 25: of the same order noise: of the same order as the noise
*This was changed accordingly in the main text.*

Pg 2 – Ln 26: which limits: complicating

This was changed accordingly in the main text.

Pg 2 – Ln 30: 10-day glacier velocity changes over 7 years: I know what is meant, but found this notation somewhat confusing. Later on it is also mentioned that 20-day time steps are used. Consider rephrasing, also elsewhere.
This was rephrased. It now reads:
We analysed thousands of images and retrieve, for 7 years, velocity changes over 10-day intervals for 38 glaciers in the Pamir.
...

Pg 2 – Ln 42: compared to noise: compared to the noise
This was changed accordingly in the main text.

Pg 2 – Ln 43: velocity changes over 48 glaciers: velocity changes for 48 glaciers
This was changed accordingly in the main text.

Pg 2 – Ln 45: Both result from changes in meltwater input: Wording too strong, 'both appear to result'
This was changed accordingly in the main text.

Pg 3 – Ln 50: future changes: future change
This was changed accordingly in the main text.

Pg 3 – Ln 58: poor spatial coverage: poor spatial and temporal coverage
This was changed accordingly in the main text.

Pg 3 – Ln 63: Consider adding: T. Strozzi, A. Luckman, T. Murray, U. Wegmuller and C. L. Werner, "Glacier motion estimation using SAR offset-tracking procedures," in IEEE Transactions on Geoscience and Remote Sensing, vol. 40, no. 11, pp. 2384-2391, Nov. 2002, doi: 10.1109/TGRS.2002.805079.
This was changed accordingly in the main text.

Pg 3 – Ln 72-73: The references seem to be provided as examples of ice velocity derived from images with a large time interval (annual to multi-year time periods) to support the previous sentence "The time interval … of the measurement.", but note that in at least some of these references (e.g. Rignot) this is not the case. In general for SAR imagery a shorter time span is advantageous due to better coherence.
We specified that large time intervals are better especially for optical images:

*The time interval between optical images that are correlated is an important parameter, as a larger time interval will increase the signal (i.e., the displacement) relative to the noise of the measurement.*

Pg 4 – Ln 93-98: New generations…: Worth mentioning here is also the crucial role of systematic acquisition planning.
This is now mentionned.

Systematic acquisition planning and new generations of medium-resolution, short-recurrence-time optical sensors

Pg 5 – Ln 112: recovered: retrieved
This was changed accordingly in the main text.

Pg 5 – Ln 117: meteorologic: climatic (as in your title)
This was changed accordingly in the main text.

Pg 8 – Ln 153: for the time periods above: mention the time period here again.
This was changed accordingly in the main text.
We downloaded Landsat 8 and Sentinel 2 images that cover our study area for the time period 2013-2020.

Pg 8 – Ln 163: ESA Sentinel-2: Copernicus Sentinel-2
This was changed accordingly in the main text.

Pg 9 – Ln 187: we used the quality assessment band: it is not mentioned how this band is used, presumably as to mask the flagged pixels. Please explain.
This was changed accordingly in the main text.

*We used the quality assessment band for Landsat 8 images to flag and remove pixels where clouds*

Pg 12 – Ln 244: for each pixel of all measured velocities: for each pixel and of all measured velocities
This was changed accordingly in the main text.

Pg 13 – Ln 257-273: Was there any longer term velocity trend for any of the glaciers and did you remove this? If not, could this affect your results?
We did not observed clear long-term trend, as also supported by the study of Dehecq et al., 2019.
We precised it to:
In agreement with Dehecq et al., (2019) we did not observe any clear long-term trend in the velocities that we could correct for.

Pg 14 – Ln 289: changes in the velocity: changes in the velocity relative to the multi-annual mean
This was changed accordingly in the main text.

Pg 14 – Ln 290: spring 2013 to winter 2019-2020: if I read these graphs correctly, they seem to run from spring 2013 to winter 2020-2021 (end of 2020). Please check.
This was changed accordingly in the main text.

*From spring 2013 to winter 2020-2021*

Pg 14 – Ln 292-293: Such changes are higher than those described in the study of Lambrecht et al. (2014): How much higher?
This was changed accordingly in the main text.

*Such changes are two to three times higher than those described in the study of Lambrecht et al. (2014).*

Pg 14 – Ln 299-300: likely because of the combination of Landsat 8 and Sentinel-2 data:

it would perhaps be interesting to mention if you found any differences in measured velocities between L8 and S2 for the same time period.

In line with Millan et al., 2019 we did not find important changes between the two datasets when considering the same time span. The combination provides a better temporal coverage and therefore allows to have a more robust estimation of the velocity field.

Pg 17 – Ln 348: H765Glacier: H765 Glacier
This was changed accordingly in the main text.

Pg 19 – Ln 378: 20 glaciers: Please check, in Figure 6 and several other places in the manuscript you mention 24.
We checked and corrected where needed, this is indeed 24 glaciers

Pg 19 – Ln 386: created: created
This was changed accordingly in the main text.

Pg 23 – Ln 458: Figure 8c: Describe figures in the correct order (first 8a & 8b).
This was changed accordingly in the main text and the two sentences are now switched

Pg 23/24 – Ln 464/465: there is no clear influence of the glacier velocity on the downglacier migration rate: 1) add reference to figure 8e; 2) maybe it is a trick of the eye, but to me there appears to be a negative correlation.

There is indeed a slight negative correlation. This is now mentioned in the text:

*When normalizing the migration rate with the slope of the glacier, the downglacier migration rate appears to be negatively correlated with the glacier velocity (Figure 8e).*

Pg 28 – Ln 538: describes: follows
This was changed accordingly in the main text.

Pg 30 – Ln 586: (-0.05 m.d-1, -18.25 m.year-1): why negative if you talk about accelerations?
This was a typo which has now been corrected.

Pg 32 – Ln 643/646: the links do not work
It should work now.

Figures:
The figures and additional video supplement are nice and very informative, some minor comments below:

Fig 1: Caption: "mountain range area": "mountain range"
This was changed as suggested.

Fig 1: Caption: "m.d-1" : "m d-1" also elsewhere in manuscript.

This was changed as suggested.

Fig 1: Caption: linear scale going from black (no displacement) to white (fast displacement): the scale seems to go from yellowish white to blueish white, but not from black to white.

This was changed as suggested.

Fig 1: Caption: background elevation: background shaded relief

This was changed accordingly in the text.

Fig 1: Caption: '(c): (c)

This was changed accordingly in the figure.

Figure 3: 3b) Add label for vertical axis

This was changed accordingly in the figure.

Figure 3: Consider making c & d the same size.
We extended the panel (d) in order to give a better picture of the dynamic since. Making both panel the same size would decrease the readability of the figure.

Figure 4: The color scale mentions m.d-1, I assume this must be %

This was changed accordingly in the figure; it was indeed in %

Figure 4: Add label to y-axes
The units (km) refer to the elevation and the (m/d) to the velocity. We specified it in the caption

Figure 4: Some numbers are partly obscured (see for example 4d x-axis)
We modified the figure to correct this

Figure 4: 4f) W731 seems to show up to 5/6 speed-up events, are these real, could you elaborate?
Figure 5: Add label to y-axes
The label is the month of year as already shown

Figure 6: n=24: see previous point regarding comment on Pg 19-Ln 378.
This was checked in the text and indeed it is 24 glaciers, and not 20 as mentioned on line 378